# Study of the Transcription Effects of Pressing Dies with Ultrasonic Polishing on Glass Molding

**Ken-Chuan Cheng [1], Chien-Yao Huang [2], Jung-Chou Hung [3], A-Cheng Wang [4],* and Yan-Cherng Lin [5]**

1. Department of Mechanical Engineering, Chung Yuan Christian University, Taoyuan City 320, Taiwan; kccheng20@gmail.com
2. Taiwan Instrument Research Institute, National Applied Research Laboratories, Hsinchu City 300, Taiwan; msyz@narlabs.org.tw
3. Department of Mechanical Engineering, National Central University, Taoyuan City 320, Taiwan; hungjc@ncu.edu.tw
4. Department of Mechanical Engineering, Chien Hsin University of Science and Technology, Taoyuan City 320, Taiwan
5. Department of Mechanical Engineering, Nankai University of Technology, Nantou City 542, Taiwan; ycline@nkut.edu.tw
* Correspondence: acwang@uch.edu.tw; Tel.: +886-3-4581196 (ext. 5527)

**Abstract:** The micro lens array (MLA) has played an important role in optical systems for the past few years, and the precision of pressing dies has dominated the quality of MLAs in glass molding. Few studies have covered the transcription effects on surface roughness of pressing dies for this technology. Therefore, this research utilized pressing dies to produce a sine-wave lens array on glass molding, to transform the Gauss-distributed spotlight into a uniform straight one and then characterize the transcription effects of these lenses. Pressing dies with a sine-wave shape were firstly cut by wire electrical discharge machining (WEDM), and then ultrasonic polishing using diamond abrasives was applied to finish the sine-wave surface with an original roughness of 0.2 μm Ra. Next, the sine-wave lens arrays were pressed by glass molding at the appropriate pressure and temperature, before evaluating the transcription effects of transforming the Gauss-distributed spotlight into a uniform straight one. The result showed that the sine-wave lens array stuck easily to the pressing die and then ruptured during glass molding due to the poor surface roughness of pressing tool. However, the diamond abrasive with appropriate sizes could establish good surface roughness on pressing dies via ultrasonic polishing, and the pressing die with a low surface roughness of 0.08 μm Ra was able to successfully perform MLA in the glass molding. However, only pressing dies with a surface roughness smaller than 0.023 μm Ra could produce precision glass lenses to transform the Gauss-distributed spotlight into a uniform straight one.

**Keywords:** micro lens array; glass molding; ultrasonic polishing; transcription effect; surface roughness





## 1. Introduction

In recent years, the micro lens array (MLA) has become a vital part of optical systems [1,2]. One of the main applications of MLAs is to make the distribution of light energy uniform, and shapes of the lens are applicable to precision measurements, calibration, robot visual systems, light source control of optical systems, etc. Generally speaking, precise metal lenses and glass lenses are usually made using the diamond grinding process [3]; however, the grinding machine is very expensive and has high requirements in terms of operators and the environment. In addition, the long grinding time in making precise lenses is also a high manufacturing cost. As far as the quality in the course of production is concerned, glass molding technology is an efficient means of producing MLAs [4,5]. The precise glass molding (PGM) technique is an ideal processing method for fabricating high-precision glass elements because of its short cycle time, high accuracy, and high freedom;

moreover, ultrathin PGM is a very important technology to fabricate high-precision glasses for application in 2.5D/3D smartphones [6]. In order to meet the demand for high-precision MLA, the precision of the pressing dies in the PGM method is of great importance [2–4]; however, the shapes of dies, in general, are too complicated to polish.

Ultrasonic polishing uses high-frequency vibration of the tool combined with a reciprocating motion to push the abrasives to abrade the surfaces of the workpiece with small forces, and a smooth and precise surface can be obtained [7,8]. Yan et al. proposed a machining method that combines micro electrical discharge machining and ultrasonic vibration polishing to create high-precision micro holes with high aspect ratios [7]. This study indicated that the micro hole surface could be lapped effectively by the abrasives coupled with a rotating and feeding micro tool. Kobayashi et al. [9] developed a novel polishing apparatus by making a polishing pad to create an elliptic ultrasonic vibration, which induced the abrasives held by the pad to impact the work surface at extremely high acceleration. Therefore, a high material removal rate and good surface quality could be achieved, even at a low rotational speed of the wafer or small flow of slurry. Tsai et al. [10] proposed chemical mechanical polishing with ultrasonic vibration to finish a copper substrate, resulting in a passive layer on the copper surface, formed by the chemical action of the slurry, which could be removed not only by the mechanical action of chemical–mechanical polishing but also by ultrasonic action. In consideration of high-hardness and brittle materials such as glasses and ceramic, traditional grinding methods are not suitable [11,12]. Zhao [11] put forward an abrasive polishing method with ultrasonic vibration to finish the micro cylindrical surface of SiC, aiming to enhance the surface quality and improve the machining efficiency. Experimental results showed that the low polishing force, low rotational speed, high vibration frequency, and high amplitude could result in a low surface roughness and fewer polishing marks on the micro cylindrical surface. Furthermore, the cylindrical arrays of SiC were successfully polished during the optimized parameters using a precision diamond as the working tool. Yu [13] applied ultrasonic polishing with an atomizing liquid to finish an optical glass lens. A nozzle was applied in the machining process to inject the atomizing liquid into the working area to decrease the abrasive clustering effects, improving the surface quality in the polishing process. The result showed that the surface roughness of the optical lens could reach 2 nm during this machining method. Moreover, a mathematic model was set up by Li et al. [14] to interpret the increase in material removal rate for vibration process. The results of the model showed that the vibration could improve material removal by increasing vibration amplitude in the vertical direction; however, the horizontal vibration only contributed little to the material removal rate, which agreed well with experimental results.

Since dies and molds are very important in PGM, for finishing molds with a high numerical aperture, Suzuki et al. [15,16] developed an ultrasonic vibration polishing machine driven by piezo-electric actuators. They found that the surface roughness of micro-aspheric molds of tungsten carbide could be reduced to 8 nm Rz. Since ultrasonic polishing performs exclusively well to obtain a smooth and precise surface of an element, this study aimed to utilize a polishing rod with ultrasonic vibration and a reciprocating motion to enhance the accuracy of pressing dies with a sine-waved shape. Furthermore, the transferring effects of sine-wave dies on glass molding were used to evaluate the surface roughness of the pressing die. The object of this research was to apply the ultrasonic machining technology to replace diamond grinding to create a precise optical lens mold.

## 2. Methods

### 2.1. Optical Design

Sine-waved lenses, which possess the function of transforming a Gauss-distributed spotlight into a uniform straight line, were designed in this research. The equation of the sine wave of the pressing die is as follows:

$$Z(t) = A \times \sin(2\pi f t), \tag{1}$$

where $A$ is the amplitude, $f$ is the ordinary frequency, and $\omega = 2\pi f$ is the angular frequency. The parameters in this investigation were as follows: $A = 100$ μm, $f = 1$, and $\omega = 6.283$. Figure 1 shows the result of the optical design. The spotlight was transformed into a line shape, and the intensity was near-uniform through the sine-waved lens.

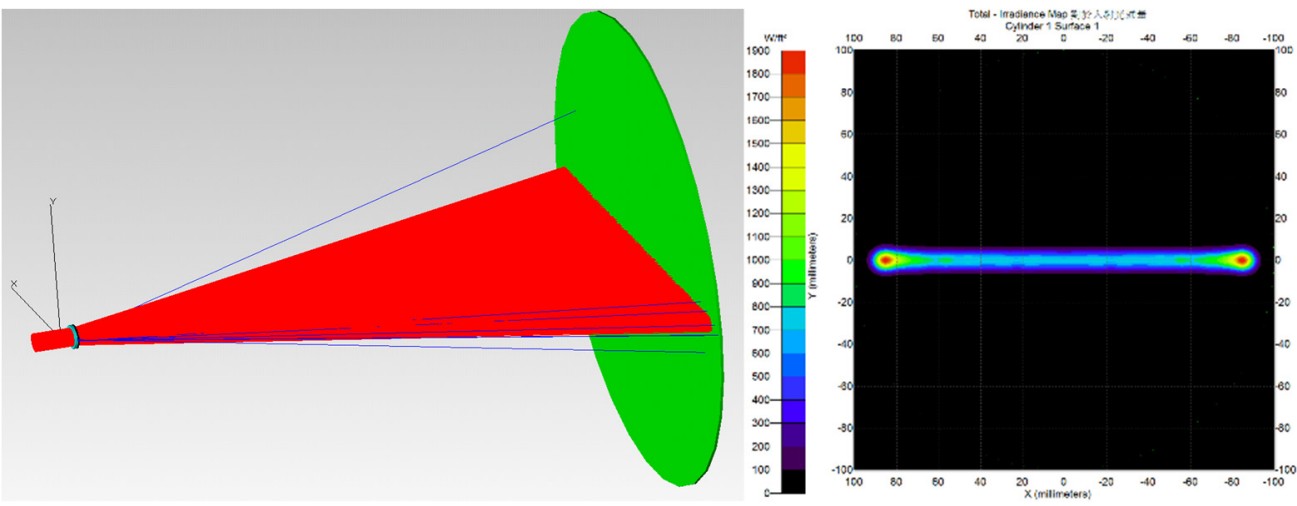

**Figure 1.** Optical design of lens for transforming Gauss-distributed spotlight into a uniform straight line.

### 2.2. Pressing Die of Glass Molding

The main purpose of the experiment was to demonstrate the effect of a sine-waved MLA on the uniformity of light energy. During the experiment, this study first utilized the CAD software to design a pressing die with sine-waved profiles, and then cut it out using a wire electro discharge machine; due to poor precision on the surface of the pressing die after wire electro discharge machining, an ultrasonic equipment that induces high-frequency vibration (30 kHz) of the polishing rod was used to perform the polishing process. Diamond slurries were applied as polishing abrasive in the pressing die finishing process; hence, the effects of diamond powders in the polishing method are discussed here to characterize the precision of the pressing die. Moreover, a polymer rod with long fibers was utilized as the polishing tool to effectively transmit the ultrasonic vibration in the machining process. Figure 2 presents a diagram of the ultrasonic polishing process. The degree of surface roughness depended highly upon the polishing path and working time. Figure 3 shows the polishing path of the ultrasonic tool when using the CNC machine. The polymer rod followed circular and reciprocating paths created by CAD/CAM software to finish the sine-wave surface of the pressing die; in the experiment, three values of surface roughness (0.20 μm Ra, 0.08 μm Ra and 0.023 μm Ra) were utilized to make the MLA from PGM. On the basis of the results, the transcription effect of the lens and the precision of MLA were investigated in this research.

### 2.3. Material

SKD-11 mold steel with the size 15 mm × 15 mm × 15 mm was chosen as the pressing die in PGM; this material is usually used to create molds and dies in the molding industry. Moreover, soda lime glass with a diameter of 10 mm and thickness of 1 mm was chosen as the PGM material; this glass has excellent light transmittance, an even surface, and low thickness. The composition of the glass is 73% $SiO_2$, 13% $Na_2O$, 10% $CaO$, 3% $MgO$, and 1% $Al_2O_3$, and it is often applied as a component of LCDs. The physical properties of soda lime glass are listed in Table 1.

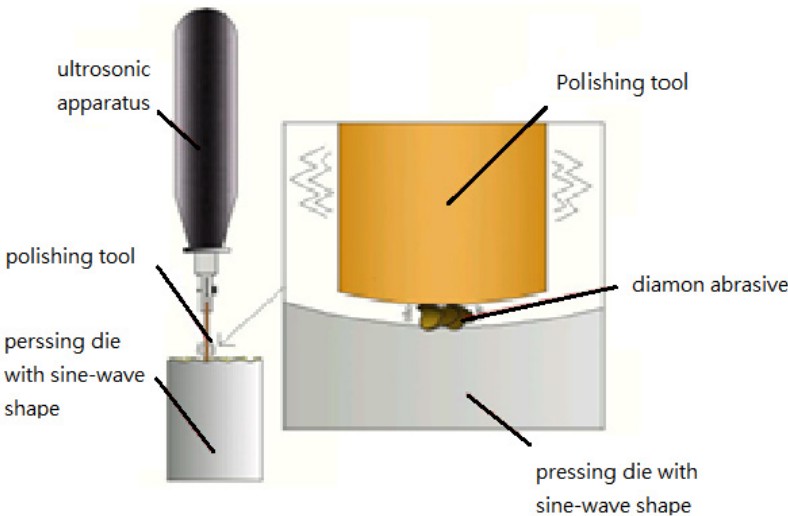

**Figure 2.** Diagram of the ultrasonic polishing process.

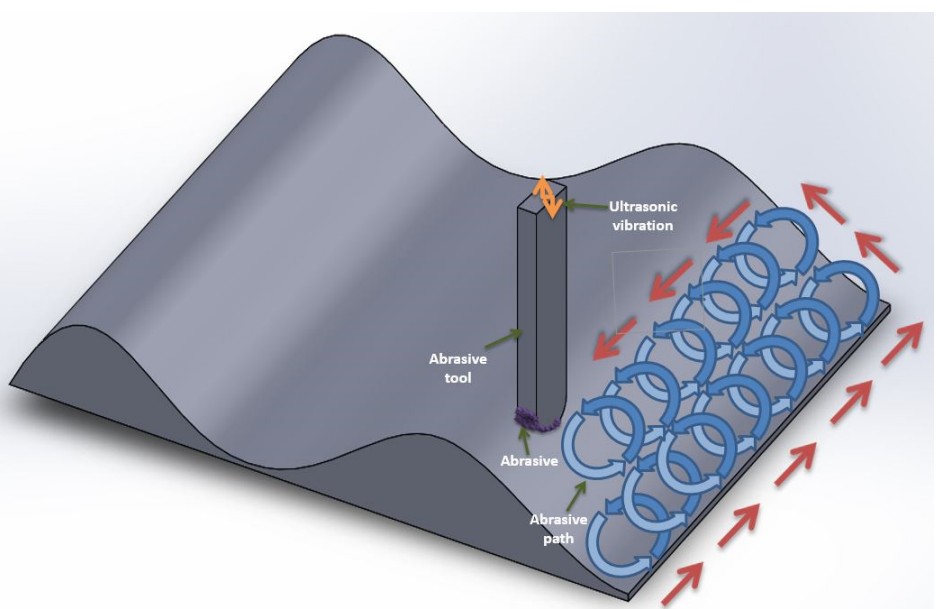

**Figure 3.** The ultrasonic polishing path followed by CNC machine.

**Table 1.** The physical properties of soda lime glass.

| Physical Properties | Parameters |
|---|---|
| Glass transition temperature, Tg (°C) | 564 |
| Thermal expansion coefficient (ppm/K) | 9.5 |
| Refractive index at 20 °C (nD) | 1.52 |
| Dispersion at 20 °C, 140 × (nF − nC) | 87.7 |

### 2.4. Glass Molding Process

In this study, the procedure of producing MLA in the glass molding involved four steps. Firstly, a preformed circular disc of glass 10 mm in diameter was placed on the mold core located in the chamber to prepare for the glass molding process. Then, the chamber was pumped in a vacuum state to avoid gas impurities, which would affect the quality of the micro lens in the molding process. In addition, to easily form the MLA, the preformed circular disc and mold core were heated to 680 °C, before being maintained to stabilize

their temperature. Finally, a pressing force exceeding 210 N was applied to the top- and bottom-mold cores to press the circular glass disc into a sine-waved shape; however, MLA maintained the same pressure for a certain time in the chamber and was cooled by nitrogen before removing the element from the chamber. Figure 4 shows the hot-pressing stage of glass molding.

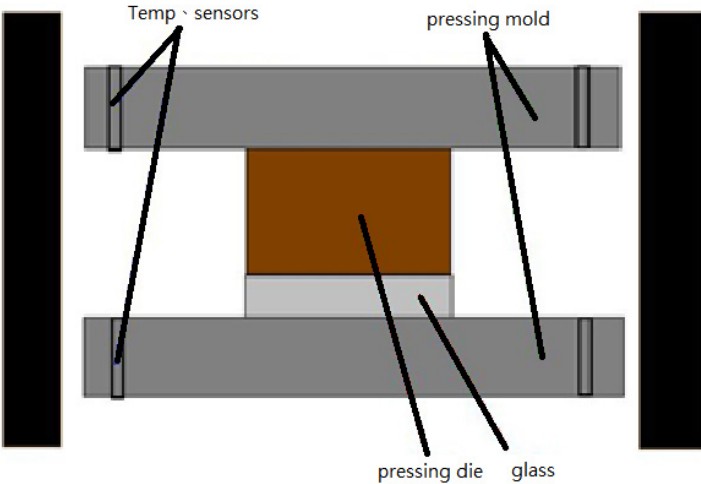

**Figure 4.** Hot-pressing stage of glass molding.

*2.5. Evaluating the Precision of Pressing Dies and Transcription Effects in the Glass Molding*

In order to fully understand the transcription effects of the surface roughness of pressing dies on the MLA surface topography during the glass molding, this study firstly investigated the surface roughness of the sine-waved shapes after ultrasonic polishing. The roughness improvement ratio (RIR) is defined in Equation (2) in conjunction with the surface roughness of pressing dies before and after polishing, which is the ratio of the difference in surface roughness before and after polishing. Furthermore, considering that the quality of the MLA after glass molding is strongly relevant to the surface roughness of the pressing die, and to understand the effects of the pressing die on the surface roughness of MLA during the process of glass molding, the transcription ratio (Tr) is defined in Equation (3), which is the surface roughness of the glass after PGM with respect to that of pressing dies during polishing. A flowchart of this research is provided in Figure 5.

$$\text{RIR} = \frac{\text{Ra}_{\text{inital}} - \text{Ra}_{\text{final}}}{\text{Ra}_{\text{inital}}} \times 100(\%). \tag{2}$$

$$\text{Tr} = \frac{\text{Ra}_{\text{glass}}}{\text{Ra}_{\text{die}}} \times 100(\%). \tag{3}$$

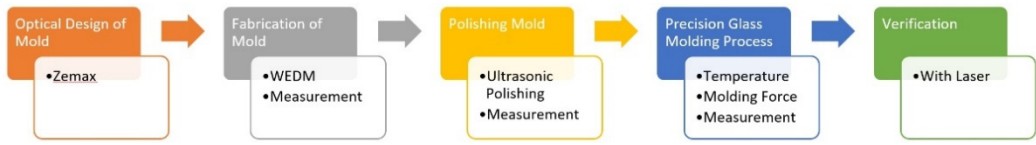

**Figure 5.** Flowchart of the experimental procedure to make the MLA.

## 3. Results

*3.1. The Transcription Effects of Temperature and Pressing Force on the Surface Roughness of MLA*

Temperature and pressure are the key parameters of the glass molding. Tr was evaluated herein for the surface roughness of pressing dies and the surface roughness of MLA after PGM. In general, a good transcription effect was obtained if the surface roughness of the glass lens and the pressing die were the same. Figure 6 presents the effects of temperature on the surface roughness of the pressing die and MLA; the results show

that the surface roughness of the pressing die was 0.045 μm Ra and the surface roughness of the MLA was 0.026 μm Ra, the Tr of the pressing die to the glass lens could only reach a low value of 58% at a temperature of 680 °C, and the surface morphology of the pressing die could not totally reflect that of the glass surface, thus inducing a bad transcription effect at this temperature. However, Tr could obtain a high value of 99% at a temperature of 690 °C, and the glass lens did not stick to the die or become crushed after PGM. Moreover, the glass products could stick to the pressing die at temperatures over 690 °C in PGM. Therefore, a temperature of 690 °C was utilized as the working parameter to make the MLA. Figure 7 shows the glass molding surfaces at different temperatures. Figure 7a presents the surface deformation at 680 °C; the result shows that the surface morphology of the pressing die was not totally transcribed to the glass surface. However, the appearance of the pressing die could be effectively transcribed to the glass after molding in Figure 7b. Figure 8 displays the effects of pressure on the surface roughness of the pressing die and MLA; the results show that the surface roughness of the pressing die was 0.042 μm Ra while that of the MLA was 0.033 μm Ra, Tr of the pressing die to the glass lens was only 78.6%, and the surface morphology of the pressing die also could not be fully transferred to the glass surface at a force of 210 N. However, Tr could get an excellent value of 98% at a force of 250 N, whereby the MLA did not stick to the pressing die. However, the MLA was ruptured when a force of 270 N was applied. Thus, 250 N was chosen as the pressing force to make the MLA. Figure 9 shows the glass appearance after glass molding; the results display that the surface morphology of the pressing die could not be fully transcribed to the glass surface at 210 N, whereas it was well transcribed at 250 N.

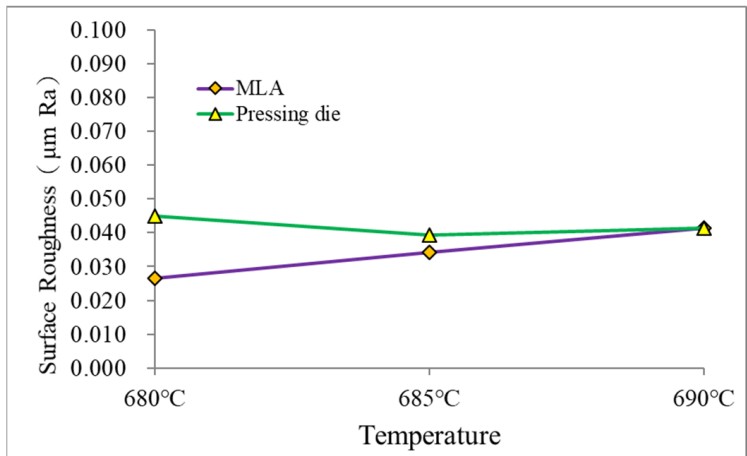

**Figure 6.** Effects of temperature on surface roughness between pressing die and MLA.

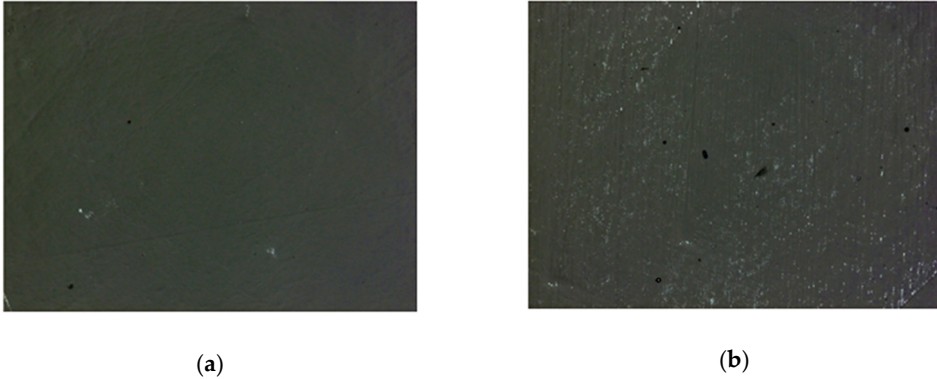

(**a**)          (**b**)

**Figure 7.** SEM diagrams at different molding temperature: (**a**) 680 °C; (**b**) 690 °C.

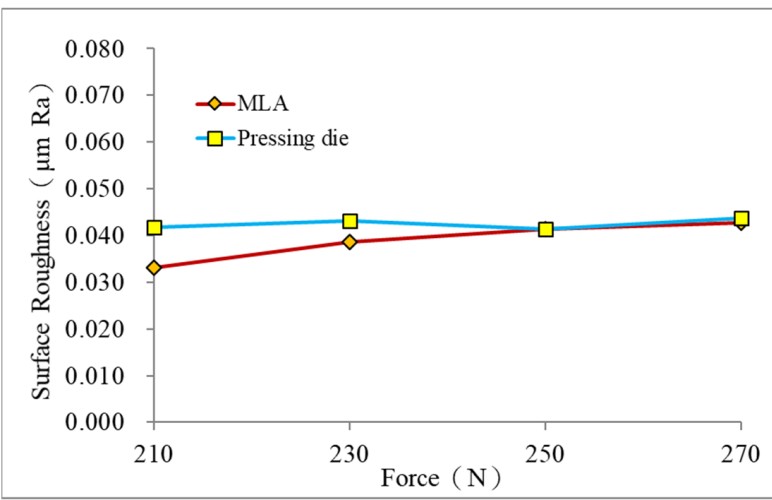

**Figure 8.** Effects of pressing force on surface roughness between pressing die and MLA.

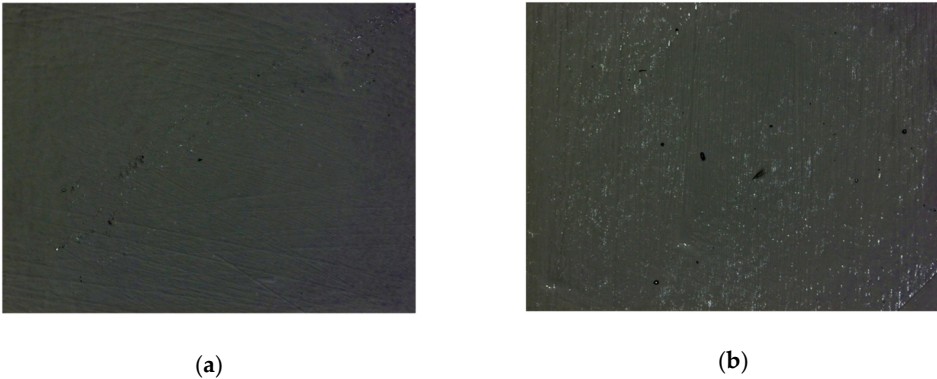

|(**a**)|(**b**)|

**Figure 9.** SEM diagrams at different molding pressure: (**a**) F = 210 N; (**b**) F = 250 N.

*3.2. Polishing Effects with or without Ultrasonic Polishing on Surface Roughness and Material Removal*

By virtue of the fact that it is difficult to polish the surface of pressing dies with sine-waved shapes, in order to obtain a relatively smooth and precise surface, an efficient and feasible technique is represented by ultrasonic polishing using the high-frequency vibration of the tool combined with a reciprocating motion. In this experiment, the ultrasonic polishing method was applied to investigate the finishing effects of the surface roughness on the pressing dies. For the purpose of comparison, the pressing die was firstly polished without ultrasonic vibration according to the motion paths in Figure 3 using a polymer rod with long fibers; the next step was to utilize the polishing tools with ultrasonic vibration to perform the same machining process as the first step on the surface of the pressing die. Diamond powders of mesh no. #8000 were utilized as the abrasive in both experiments. Figure 10 exhibits the corresponding results, showing that, with or without ultrasonic polishing, the surface roughness of the pressing die was decreased with increasing machining time, whereas the die surface with ultrasonic polishing was smoother and produced a larger amount of material removal than the die surface without ultrasonic finishing. This phenomenon shows that, by means of high-frequency vibration combined with the complex motion of the polishing rod, diamond abrasives could burnish the sine-waved surfaces of the pressing die to yield a very smooth morphology. As also shown in Figure 10, the surface roughness of the pressing die decreased from 0.2 um Ra to 0.023 μm Ra when using the polishing rod with ultrasonic vibration after 25 min of machining, and the RIR of the pressing die could reach 90%; in contrast, when the pressing die was polished using only the reciprocating motion without ultrasonic vibration, the diamond abrasives could not apply an efficient finish to the die surfaces, resulting in a low RIR of

33%. Figures 11 and 12 show images of the pressing dies after the polishing processes; in Figures 11a and 12a, the obvious cut marks by WEDM are still visible on the die surfaces during the polishing process without ultrasonic vibration, whereas a smooth and precise surface could be obtained during ultrasonic vibration, as shown in Figures 11b and 12b.

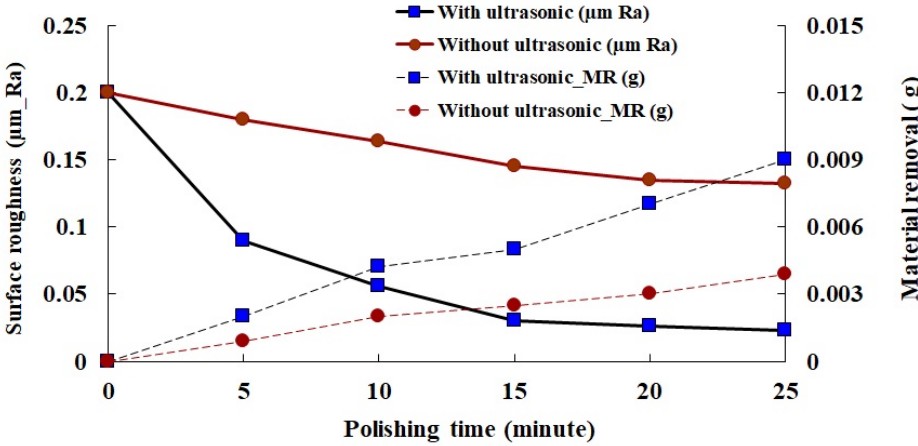

**Figure 10.** Effects of ultrasonic polishing on surface roughness and material removal.

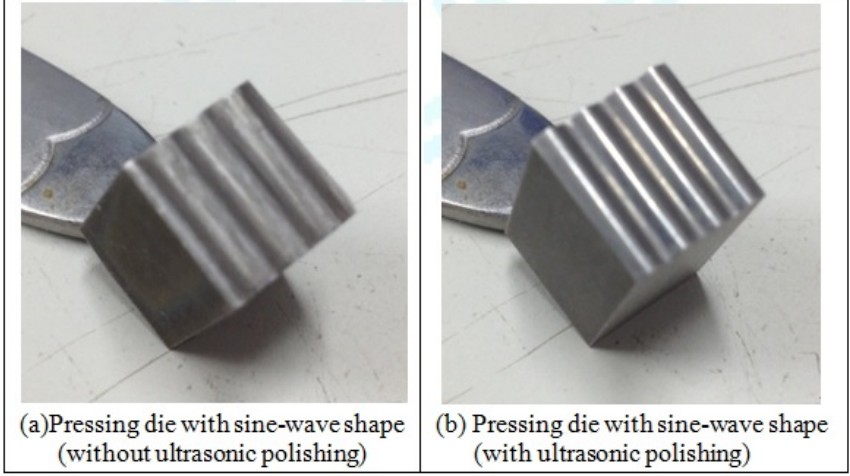

**Figure 11.** Profiles of pressing dies with or without ultrasonic polishing.

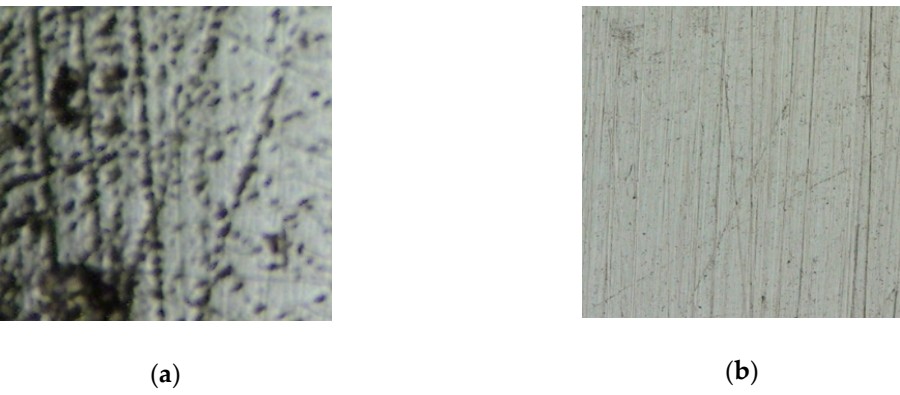

**Figure 12.** Machined surface of the pressing dies without and with ultrasonic vibration: (**a**) machined surface without ultrasonic vibration; (**b**) polished surface with ultrasonic vibration.

### 3.3. Polishing Effects of Diamond Abrasive on the Surface Roughness and Material Removal

Inasmuch as the surface roughness of pressing dies may highly influence the transcription effects on MLA, this section evaluated the polishing effects of diamond abrasive sizes on the surface roughness and material removal in the pressing dies. Figure 13 displays the effects of the mesh no. of diamond abrasives (#3000, #5000, #8000, #15000, and #28000) on the polishing of the sine wave dies under at a force of 250 N and a temperature of 690 °C. The results show that regardless of the mesh of diamond abrasive utilized in the pressing die polishing, the surface roughness of pressing dies decreased with an increase in polishing time; furthermore, the material removal increased with an increase in working time. Generally, a small mesh number of the abrasive represents a large grain size, which produce deep and obvious abrading marks on the working surface; therefore, high material removal can be easily achieved. The results of Figure 9 also show that the abrasive of mesh #3000 could achieve a better surface roughness than the other meshes over 10 min, due to the large grains of the abrasive quickly removing the recasting layers created by WEDM. However, since these large grains of the abrasive could also make deep abrading marks during ultrasonic polishing, the surface roughness would quickly reach a saturated level. Therefore, the abrasive of mesh #8000 could create a higher surface roughness than after 25 min of machining. The results show that the best RIR (89%) of the sine wave die was achieved, with the surface roughness decreasing from 0.20 μm Ra to 0.023 μm Ra, using the abrasive of mesh #8000 after polishing for 25 min. In contrast, the surface roughness of pressing dies could not achieve fine values when the abrasives of mesh #15000 and #28000 were used due to their small grain sizes, whereby only tiny forces were applied to the polishing die surface without the ability to effectively remove the recasting layers, thus inducing poor material removal and surface roughness after polishing for 25 min. Figure 14 presents the fine surface profiles of the sine-waved glass and the MLA produced using this pressing die. The die surface became very smooth and very bright when the surface roughness was decreased to 0.023 μm Ra (Figure 14a), while a good shape of MLA was produced using this pressing die (Figure 14b).

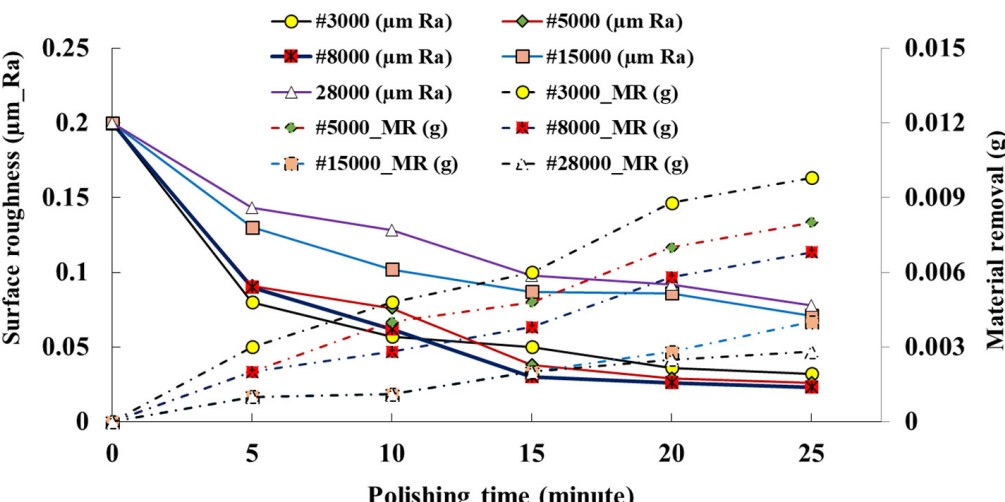

**Figure 13.** The effects of abrasive size on surface roughness and material removal.

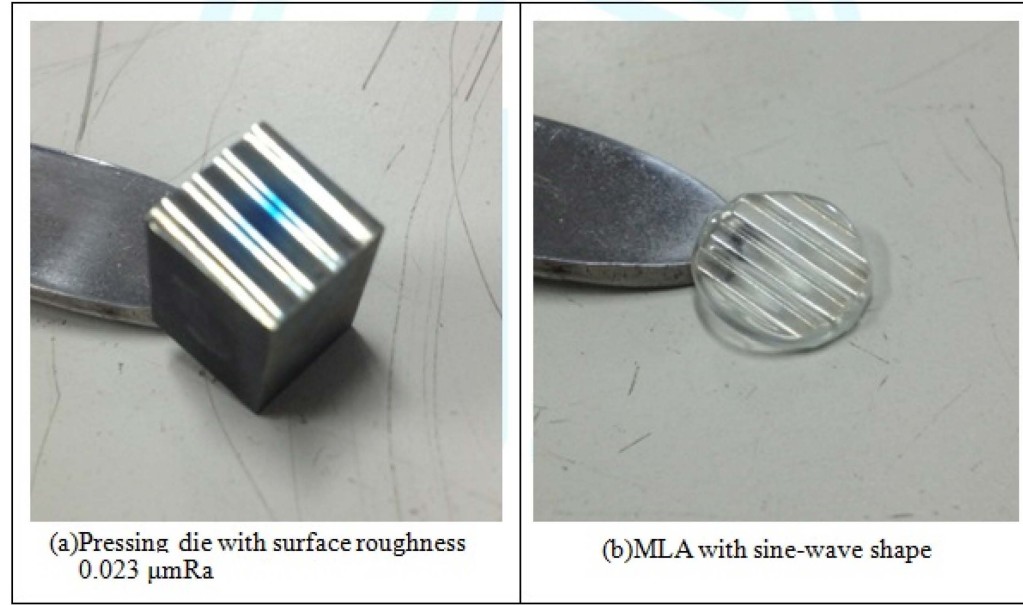

**Figure 14.** Pressing die with sine-wave shape (surface roughness 0.023 μm Ra) and MLA produced using this die.

### 3.4. Transcription Effects of the Pressing Dies

The study applied the pressing dies after ultrasonic polishing to produce MLAs with sine-waved shapes, such that the relationship between the transcription effects of the pressing dies and the MLA could be evaluated. Figure 15 presents the shape of the MLA with a sine wave obtained using the pressing die with a surface roughness of 0.20 μm Ra, showing that the glass lens stuck easily to the pressing die and was then ruptured after PGM. However, the MLA with a sine-wave shape was fully formed by the pressing dies with a surface roughness of 0.08 μm Ra and 0.023 μm Ra, as displayed in Figure 16. The results reveal that, using the pressing die with surface roughness of 0.08 μm Ra, the MLA with a sine-wave shape could successfully be produced, whereas the transparency of the lens was poor compared to the MLA made using the pressing die with a surface roughness of 0.023 μm Ra. Therefore, high-precision pressing dies could be used to fabricate MLAs with an excellent shape and good transcription effects in PGM; the surface roughness and Tr of the MLA were respectively 0.023 μm Ra and 99% when using the pressing die with a surface roughness 0.023 μm Ra, whereas the surface roughness was 0.074 μm Ra when using the pressing die with a surface of 0.008 μm Ra, yielding a Tr of only 92%. Figure 17 presents the results of the transcription effects of MLA after PGM using a laser emitted on the lenses. As shown in Figure 17a, the MLA made using the pressing die with a surface roughness of 0.08 μm Ra, was unable to transform the Gauss-distributed spotlight into a uniform straight line by virtue of its poor transparency. However, the MLA made using the pressing die with a surface roughness of 0.023 μm Ra in PGM resulted in a uniform straight line being obtained from the Gauss spotlight emitted by the laser pen, as indicated in Figure 17b. Thus, the high transformation effects of the pressing die with a good roughness resulted in excellent transparency of the MLA, thus enhancing the application of these glass lenses in glass molding.

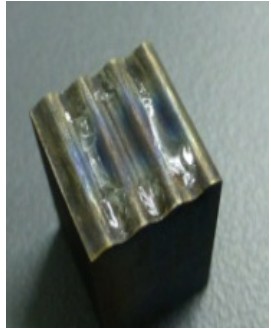
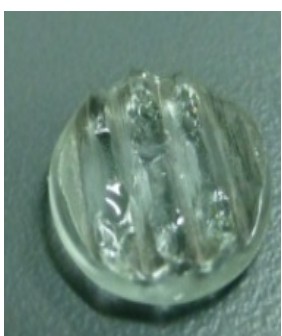

**Figure 15.** Glass stuck on the pressing die (**left**) and seriously ruptured MLA (**right**) when using the pressing die with a surface roughness of 0.2 μm Ra.

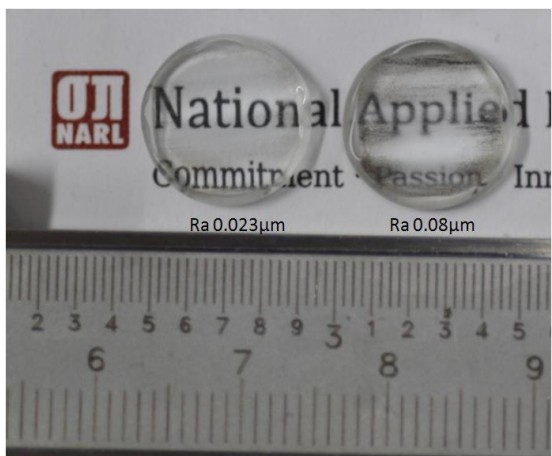

**Figure 16.** The appearances of glass MLA after PGM; the glass lens on the right was made using a pressing die with a surface roughness of 0.08 μm Ra, while the glass lens on the left was made using a pressing die with a surface roughness of 0.023 μm Ra.

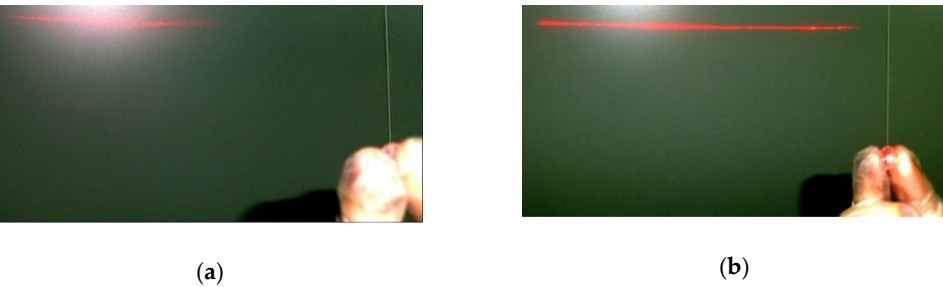

(**a**)      (**b**)

**Figure 17.** Transcription effects of pressing dies after PGM according to a laser being emitted on the lens: (**a**) MLA made using a pressing die with a surface roughness of 0.08 μm Ra; (**b**) MLA made using the pressing die with a surface roughness of 0.023 μm Ra.

## 4. Conclusions

This study created a precision pressing die with a sine-waved shape by ultrasonic polishing to produce a good-quality MLA. In this research, the surface roughness and transcription effects of the pressing die could be obviously improved, which produced a precision glass lens able to transform a Gauss-distributed spotlight into a uniform straight line. The main conclusions of this study are summarized as follows:

1.  Excellent transcription effects could be obtained when the parameter of the pressing force was 250 N and the working temperature was set to 690 °C, whereby the transcription ratio of the pressing die topography to the MLA surface could reach 99% in PGM.

2. The die surface with ultrasonic polishing was smoother and could produce a larger amount of material removal than the die surface without ultrasonic finishing. According to the results, the RIR with ultrasonic polishing of the pressing die could reach 90% when using the abrasive of mesh #8000, whereas the RIR without ultrasonic finishing of the pressing die only yielded a low value of 33%.

3. The MLA made using the pressing die with a surface roughness 0.08 μm Ra could not transform the Gauss-distributed spotlight into a uniform straight line by virtue of its poor transparency. However, the MLA made using the pressing die with a surface roughness of 0.023 μm Ra in PGM achieved a uniform straight line from the Gauss spotlight emitted by the laser pen.

4. The temperature and pressure are the key parameters of the glass molding. In this study, the MLA stuck easily to the pressing die at a high temperature (over 690 °C), whereas a low temperature and low pressure in PGM could induce a bad transcription effect, and the MLA was ruptured at high pressure (270 N).

**Author Contributions:** Conceptualization, C.-Y.H. and A.-C.W.; methodology, A.-C.W.; validation, J.-C.H., K.-C.C. and Y.-C.L.; investigation, A.-C.W. and K.-C.C.; resources, C.-Y.H.; data curation, J.-C.H. and K.-C.C.; formal analysis, K.-C.C. and Y.-C.L.; writing—original draft preparation, A.-C.W. All authors read and agreed to the published version of the manuscript.

**Funding:** This research received no external funding.

**Institutional Review Board Statement:** Not applicable.

**Informed Consent Statement:** Not applicable.

**Data Availability Statement:** MDPI Research Data Policies.

**Acknowledgments:** The authors would like to thank the Instrument Technology Research Center, National Applied Research Laboratories, Taiwan for providing the glass molding machine.

**Conflicts of Interest:** The authors declare no conflict of interest.

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
