# Peer review of "Study of the Transcription Effects of Pressing Dies with Ultrasonic Polishing on Glass Molding"

_processes, doi:10.3390/pr9112083_

Round 1

Reviewer 1 Report

The authors present an interesting work, but it must be corrected before its publication in the journal.
The references collected in the bibliography are scarce. It is not indicated how this work can contribute to the development of the SDGs and their implications for industry. Targets are fuzzy or missing.
Equation 1 is basic and nothing new is highlighted, also do not use * for multiplication, use the mathematical point symbol. The frequency is usually indicated in Greek letters.
The methods need to be further described, perhaps it helps to make an explanatory flow chart. Also, the figures are of poor quality. Remember that the International System of Units must be used (line 110, KHz - kHz)
Section 2.3 .: How is the composition of the glass determined?
Section 2.5. Write Equation and not Equ. Also, the percentage in the equations must be in parentheses
In general, the results do not show a discussion of them. It is not compared with other previous studies or carried out by other researchers. Units in many cases are glued to numbers. The same number of significant figures should be used.
The conclusions are a mere summary, without critical input. There are no future implications of this work.

Author Response

thanks for reviewer suggestion, all the responses by the authors are list as below. 

1. The references collected in the bibliography are scarce. It is not indicated how this work can contribute to the development of the SDGs and their implications for industry. Targets are fuzzy or missing.

Ans: thanks reviewer suggestion, authors already modified the introduction in line 40 to line 44 and line 93 to line 95 to explain how this work can contribute to the development of the SDGs and their implications for industry.

2. Equation 1 is basic and nothing new is highlighted, also do not use * for multiplication, use the mathematical point symbol. The frequency is usually indicated in Greek letters.

Ans. Thanks for reviewer’s correcting. The mark of multiplication had been modified to “x”. Due to the calculation of frequency value is adopted by ordinary frequency f in this paper, so the Greek letter of angular frequency ω was adding in the content of manuscript. In addition, Fig.1 was redrew to present the Equation 1 clearly for the designing concept.

3. The methods need to be further described, perhaps it helps to make an explanatory flow chart. Also, the figures are of poor quality.

Ans. Flow chart was indicated as figure 5 and Some figures with poor quality were already modified in the content

4. Remember that the International System of Units must be used (line 110, KHz - kHz)

Ans. The unit in line 110 had been modified.
5. Section 2.3 .: How is the composition of the glass determined?

Ans. The glass is purchased by a professional manufacturer and the composition of the glass is provided by the vendor.
6. Section 2.5. Write Equation and not Equ. Also, the percentage in the equations must be in parentheses

Ans. Thanks author for the correcting. All of them had been modified in this manuscript with new version.
7. In general, the results do not show a discussion of them. It is not compared with other previous studies or carried out by other researchers. Units in many cases are glued to numbers. The same number of significant figures should be used.

Ans. The content of results had been modified to show the detail discussions of this paper.

8. The conclusions are a mere summary, without critical input. There are no future implications of this work.

Ans. The content of conclusions had been modified to enhance the advantages and contribution of this paper.

Reviewer 2 Report

  1. The experimental results in Sections 3.1 and 3.2 of this paper are only described by data points and data curves, and the physical results of the experiment are not seen. I thinkthe experimental results of surface roughness between pressing die and MLA under different temperatures and pressures should be presented in the form of physical figures, as well as the difference between material removal with ultrasonic vibration and material removal without ultrasonic vibration.

  1. Infigure 8 , the cut marks on the surface of the die without ultrasonic polishing can not be clearly seen, so I think the figures of the surface morphology of the two pressing dies observed under the microscope should be attached, so as to make the difference of the surface quality between two pressing dies more obvious.

  1. Section 3.3 of this paper describes the effectof polishing with diamond abrasive on surface roughness and material removal. This section mentions the use of diamond abrasive with five sizes in polishing. However, this section more describes the characteristics of using #3000 mesh diamond abrasive in polishing, but the reason for finally selecting #8000 mesh diamond as abrasive in this paper is not stated, And the reason why the polishing time is 25 minutes is not explained, so I think this content should be supplemented. In addition, the physical result image after polishing with diamond abrasives of various sizes should be presented, so that the difference after polishing with these five mesh diamond abrasives can be seen intuitively.

  1. In Figure 13, the contrast effect of MLA made by two different roughness pressingdies is not clear because of the reflection of the lamp, especially the left figure, so I think a clearer figure should be used to show their difference clearly.

  1. In line 5 on page 6, the previous article talked about the low pressure in PGM, The glass element mentioned in the latter sentence willbe stuck on the surface of the die or be Whether it is under high pressure, if so, please explain.

  1. The data points and data lines in figures 5 and 6 are distinguished by different colors, while Figures 7 and 9 do not, and the almost coincident two lines are not easy to distinguish. Therefore, the data lines and data points in the figurein this paper should be distinguished by different colors.

  1. In line 54 of this paper, there should be a space between the two words "lapped effectively". In line 111,I thinkthe plural of "Slurrys" is wrong and should be "slurries". In line 208, there should be "surfaceroughness" There should be a space between the two words. The unit symbol of Celsius in line 314 is incorrect and should be"℃". The unit of surface roughness in lines 324, 326, 327 and 330 is incorrect and should be "μm"。

Author Response

thanks for reviewer suggestions, all the modified information are list as below.

1. The experimental results in Sections 3.1 and 3.2 of this paper are only described by data points and data curves, and the physical results of the experiment are not seen. I think the experimental results of surface roughness between pressing die and MLA under different temperatures and pressures should be presented in the form of physical figures, as well as the difference between material removal with ultrasonic vibration and material removal without ultrasonic vibration.

 Ans. Thanks for reviewer suggestion, figure 7 and figure 9 in the section 3.1 were added to explain the experimental results of surface roughness between pressing die and MLA under different temperatures and pressures

2. In figure 8, the cut marks on the surface of the die without ultrasonic polishing can not be clearly seen, so I think the figures of the surface morphology of the two pressing dies observed under the microscope should be attached, so as to make the difference of the surface quality between two pressing dies more obvious.

 Ans. Thanks for reviewer suggestion, figure 12 was added in the section 3.2 to show the die surfaces with and without ultrasonic polishing.

3. Section 3.3 of this paper describes the effect of polishing with diamond abrasive on surface roughness and material removal. This section mentions the use of diamond abrasive with five sizes in polishing. However, this section more describes the characteristics of using #3000 mesh diamond abrasive in polishing, but the reason for finally selecting #8000 mesh diamond as abrasive in this paper is not stated, and the reason why the polishing time is 25 minutes is not explained, so I think this content should be supplemented. In addition, the physical result image after polishing with diamond abrasives of various sizes should be presented, so that the difference after polishing with these five mesh diamond abrasives can be seen intuitively.

 Ans. The reasons for finally selecting #8000 mesh diamond as abrasive and why the polishing time is 25 minutes illustrated in the content of section 3.3.

4. In Figure 13, the contrast effect of MLA made by two different roughness pressing dies is not clear because of the reflection of the lamp, especially the left figure, so I think a clearer figure should be used to show their difference clearly.

 Ans. Since all the MLA made by this research were already missing and the glass molding machine was not in the Taiwan Instrument Research Institute, therefore, authors can’t make the new MLA again. And we do as possible as we can to modify the old photos to reduce the reflection of the lamp, as shown in figure 17.

5. In line 5 on page 6, the previous article talked about the low pressure in PGM, The glass element mentioned in the latter sentence will be stuck on the surface of the die or be whether it is under high pressure, if so, please explain.

 Ans. According to the results, the glass element mentioned will rupture under high pressure (270 N force). The content of manuscript had been modified in section 3.1.

6. The data points and data lines in figures 5 and 6 are distinguished by different colors, while Figures 7 and 9 do not, and the almost coincident two lines are not easy to distinguish. Therefore, the data lines and data points in the figure in this paper should be distinguished by different colors.

 Ans. The data points and data lines in figures 7 and 9 are changed to distinguished by different colors.

7. In line 54 of this paper, there should be a space between the two words "lapped effectively". In line 111,I think the plural of "Slurrys" is wrong and should be "slurries". In line 208, there should be "surfaceroughness" There should be a space between the two words. The unit symbol of Celsius in line 314 is incorrect and should be"℃".The unit of surface roughness in lines 324, 326, 327 and 330 is incorrect and should be "μm"。

Ans. Thanks author for the correcting. All of them had been modified in this manuscript with new version.

Round 2

Reviewer 1 Report

The revision has been carried out according to the indications

Reviewer 2 Report

Based on the reviewer's comments, the paper revised well. I think the paper can be accepted.